# Change in Growth Status and Obesity Rates among Saudi Children and Adolescents Is Partially Attributed to Discrepancies in Definitions Used: A Review of Anthropometric Measurements

**DOI:** 10.3390/healthcare11071010

**Published:** 2023-04-01

**Authors:** Essra A. Noorwali, Abeer M. Aljaadi, Hala H. Al-Otaibi

**Affiliations:** 1Department of Clinical Nutrition, Faculty of Applied Medical Sciences, Umm Al-Qura University, Makkah 21955, Saudi Arabia; 2College of Agricultural and Food Science, King Faisal University, Al-Ahsa 31982, Saudi Arabia

**Keywords:** anthropometry, obesity, Saudi Arabia, growth charts, growth standards, children, adolescents

## Abstract

Anthropometric measurements are the first step in determining the health status in children and adolescents. Clinicians require standardized protocols for proper assessment and interpretation. Therefore, this study aims to review the literature of international and Saudi national guidelines and studies previously conducted in Saudi children and adolescents to provide recommendations to establish Saudi guidelines in line with the Saudi 2030 Vision. Systematic search was conducted in several databases: Medline, PubMed, Saudi Digital Library and Google Scholar from January 1990 to January 2021. Further, 167 studies measured anthropometrics in Saudi children/adolescents; 33 of these studies contributed to the establishment/adjustment of Saudi growth charts or specific cutoffs or studied the trend of growth in representative samples or adjusted the international curves to be used in Saudis. This review warrants updating growth charts and establishing the standard cutoffs of Saudi adolescent anthropometrics to avoid over/underreporting. This review provides insights and recommendations regarding the resources that can be used to establish national guidelines in anthropometric measurements for Saudi children/adolescents. This review will help policymakers and the Ministry of Health to establish standardized protocols to be used in Saudi Arabia for anthropometric measurements that may assist in detecting malnutrition.

## 1. Introduction

Saudi Arabia launched the 2030 vision in 2016, which outlined the future goals of the country and their objectives. One of the vision’s themes is having a vibrant society by improving healthcare services and promoting a healthy lifestyle [1]. In addition, Saudi Arabia aims to increase life expectancy at birth by 6 years, from 74 years to 80 years, an average gain of 0.43 years per annum [1]. Determinants of life expectancy trends are dependent on the “cardiovascular revolution”, which combines smoking prevalence, obesity, lifestyle and related policies. These behaviour-related problems occur in Saudi Arabia and should be addressed in order to increase life expectancy at birth [2]. Therefore, reviewing the trends of growth, obesity, lifestyle changes and providing standardized growth charts and recommendations in establishing Saudi guidelines of anthropometric measurements is the first step that may help in increasing life expectancy. 

Anthropometry is from Ancient Greek ἄνθρωπος (ánthrōpos) ‘human’, and μέτρον (métron) ‘measure’ [3]. Anthropometrics are defined as “the study of human body measurements” [4]. Anthropometric measurements assess body composition by measuring muscles, bone and adipose tissue mass quantitively. The main anthropometric measurements that represent diagnostic criteria for obesity are height, weight, body mass index (BMI), skinfold thickness and body circumferences (waist, hip and limb) [5]. Regarding children, head circumference (HC), mid-upper arm circumference (MUAC) and growth charts [6] are part of the nutrition assessment and help in malnutrition diagnosis and predict growth [7]. Anthropometries are widely used in field and clinical situations due to their simplicity, portability, inexpensiveness and safety. Anthropometric measurements are also commonly used in large population studies due to their convenience [8]. Thus, anthropometric measurements are significant in preventing health risks, determining health status and improving individuals’ quality of life, which are all part of the Saudi Vision 2030. 

High body mass index (BMI) is a diagnostic criterion for overweight and obesity [5] and is one of the main reasons for increasing Saudi Arabia’s morbidity and disease burden [9]. Overweight and obesity negatively affect people’s socio-economic status due to their association with lower education attainment. Other economic consequences are the increase in health expenditure from treatments of overweight and obesity and their related comorbidities and their impact on productivity and workforce participation [10]. Consequently, anthropometric measurements and their cutoffs that are used to identify overweight and obesity are the first steps in determining their prevalence. 

The concerning increase in the global prevalence of obesity and its related morbidity and mortality marked the significance of the assessment of body size, shape and composition for health. Research efforts have progressed to understand obesity, and, despite these efforts, the global burden of obesity is increasing. This mandates the need for standard guidelines for anthropometric measurements to estimate the prevalence of overweight and obesity and predict healthcare costs and disease mortality. To ensure these measurements are taken precisely, clinicians must have access to national valid and reliable cutoff measures. 

To our knowledge, there are no standard national guidelines established for the Saudi population to evaluate anthropometric measurements. The issue is particularly important among children and adolescents, where there are several charts and definitions used to monitor children’s growth. Most recent studies on children in Saudi Arabia from 2010 to 2019 defined obesity and overweight using the US Centers for Disease Control (CDC) or the recent WHO criteria; others used the International Obesity Task Force criteria or the Saudi BMI percentile [11]. These differences in the standards used to define growth status in children (whether obesity or underweight) make it challenging to compare studies at the national level and, accordingly, to make accurate conclusions. For example, the prevalence of overweight/obesity has been reported to be lower when using the International Obesity Task Force (IOTF) definition compared to the use of definitions from the CDC and the WHO [12]. Overall, there is inconsistency in measuring and reporting of physical measurements among Saudi children and adolescents, which can be largely attributed to the lack of national guidelines.

Therefore, the aim of this review is to (1) review the literature on international and Saudi national guidelines of anthropometric measurements and studies previously conducted in Saudi Arabia. (2) The review will provide insights and recommendations regarding the resources that can be used to establish national guidelines and cutoffs in anthropometric measurements for Saudi children/adolescents that are in line with the Saudi vision 2030.

## 2. Materials and Methods

Several electronic databases were searched: Medline, PubMed, Saudi Digital Library and Google Scholar from January 1990 to January 2021 [13]. The search included the use of Medical Subject Headings (MeSH) and other key words (Appendix A). Hand searches of reference lists of retrieved articles were undertaken and studies prior to 1990 were included in Appendix A. After screening titles and abstracts from the searched databases, 724 full articles were screened for eligibility. A total of 167 Saudi children/adolescent studies measured anthropometrics (Figure 1). All studies in English language were included, even PhD theses and abstracts that provided sufficient information.

## 3. International and National Guidelines of Anthropometrics Used in Saudi Children

Anthropometric data in infants, children and adolescents are proxy measures for good health, dietary adequacy and optimum growth. There is almost a universal agreement on how to conduct the anthropometric measurements [14,15], but differences exist in defining what is normal vs. abnormal. Several organizations have established criteria or cutoffs to define normal and abnormal interpretations of the obtained anthropometrics. This section discusses the common international definitions of body measurements. BMI is a common proxy indicator for over- or undernutrition. Classification of BMI in children varies across different growth charts and will be discussed in the next subheadings. To calculate the *BMI*, the following is used:*Metric*: *BMI* = *Weight* (kg)/*Height* (m)^2^
*English*: *BMI* = *Weight* (Ib)/*Height* (in)^2^ × 703

### 3.1. World Health Organization

The World Health Organization (WHO) is one of the United Nations (UN) agencies that has significantly contributed to setting international guidelines and updating them throughout the years. It provides both the guidelines and the tools for applying them [15]. Currently, there are the several established guidelines for physical measurements that are widely used: the 2006 WHO growth charts [16], the 2007 WHO growth reference [17], WHO STEPS Surveillance Manual [18] and softwares for anthropometrics [19].

The 2006 WHO growth charts provide standard growth curves for children under the age of 5 years (Table 1). They were developed as part of the WHO Multicenter Growth Reference Study (MGRS) [15]. This study followed the growth of infants and children (0–5 years) from six cities in different countries, including one Arabic country (Oman). The participating sites were Accra, Ghana; Davis, California, United States (US); Delhi, India; Muscat, Oman; and Oslo, Norway. The MGRS was conducted between July 1997 and December 2003 and involved a longitudinal part for children from birth to 24 months with multiple followups and a cross-sectional part for children 18 to 71 months of age. Strict inclusion and exclusion criteria were placed to ensure uniform conditions across populations, such as, but not limited to, those related to socio-economic status, breastfeeding practices, introduction of complementary food, growth restriction factors, maternal smoking and birth outcomes (singleton and term). The final sample was about 882 children (428 boys and 454 girls) in the longitudinal component and 6669 children (3450 boys and 3219 girls) for the cross-sectional component.

### 3.2. The US Centers for Disease Control (CDC)

The US Centers for Disease Control (CDC) also provide growth standards for children, body measurements guidelines and interactive tools. These were based mainly on data from the National Health and Nutrition Examinations Survey (NHANES) with supplemental surveys and were collected solely from the American population [20,21]. The CDC growth reference has been widely used in other populations, including Saudi Arabian children. After the release of the 2005 WHO growth charts, the CDC adopted the WHO growth curves for children under the age of 2 years. For children 2–18 years, the CDC has its own charts. Several improvements have been made on the 2000 CDC growth charts throughout the years; however, the name “2000 CDC growth charts” remained the same [6]. 

Differences exist between the CDC and WHO growth charts depending on the age group, growth indicator and specific Z-score curve [22]. These differences were of importance during infancy, which led the CDC to adopt the WHO growth charts for those under the age of 2 years. Moreover, the WHO length/height-for-age standards have less variability compared to the CDC. Both the weight-for-length and weight-for-height charts indicated that children in the CDC (US studies) were generally heavier. This was reflected in the BMI for age, leading to major differences in the estimates of underweight, overweight and obesity; higher rates of obesity and lower rates of underweight are expected when the WHO BMI-for-age standard is used [22]. 

### 3.3. The International Obesity Task Force (IOTF) BMI Cutoffs

Other cutoffs that have been widely used in children are those established by the International Obesity Task Force (IOTF) (Table 1). Centiles were constructed for children 2 to 18 years using data obtained between 1963 and 1993 from six countries: the UK, USA, The Netherlands, Brazil, Singapore and Hong Kong. No cutoffs are available for children under the age of two years. The IOTF standards provide only BMI cutoffs for overweight, obesity and underweight (thinness) by linking the data to adult’s BMI cutoff points [23,24,25]. Standards for overweight [25] and underweight [24] were published in 2000 and 2007, respectively. Some improvements were completed in 2012 using the same data to allow for calculating the SD scores in addition to the centiles [23]; the differences between the original and the updated curves have minimal impact on BMI classifications [23,26].

### 3.4. International Diabetes Federation (IDF)

Central (abdominal) obesity is an indicator of adverse health outcomes and is commonly estimated by measuring the waist circumference. The IDF also provides waist circumference percentiles for children and adolescents starting at age 6 years, but the IDF does not recommend the diagnosis of metabolic syndrome for children under 10 years. Cutoffs used were categorized based on age groups 6 to <10, 10 to <16 and ≥16 years [27]. The IDF waist circumference cutoff for adolescents has been reported to be applicable to Spanish adolescents in predicting metabolic syndrome [28].

### 3.5. The Saudi National Guidelines

Growth reference for length/stature, weight and head circumference were published in 2007 based on a representative sample of children and adolescents from the 13 administrative regions in Saudi Arabia [29]. The sample included 35,279 eligible healthy children and adolescents from birth to 19 years of age. Charts for BMI were developed later in 2009 that appear to be completed using the same sample of children included in El-Mouzan et al. (2007) [29]; however, in the methodology, they did not refer to the original article that collected the sample [30]. For children, the Saudi BMI-for-age charts provided by the Ministry of Health (MOH) do not have percentiles below the 50th percentile, although the original article provides them [29]. The booklet provides them for <60 months: https://www.moh.gov.sa/en/Ministry/MediaCenter/Publications/Pages/Child-Health-Passport.pdf (accessed on 1 January 2022). 

The 2005 Saudi growth charts are the ones endorsed by the Saudi MOH [31,32]. However, recent modifications to the charts of children 0–60 months of age have not been adopted yet [33,34]; one re-analysed the data to reconstruct the percentiles for weight, length/height, head circumference and body mass index for age and weight for length/height [33], whereas the second one created Z-scores reference values for the same age group [34]. There have been Saudi growth standards established by the MOH for children under the age of 5 years published in 2000 [35] and 2004 [36], but they covered five regions only out of the thirteen regions and are no longer used; the two studies appear to be the same, with slight differences in authors list and the journal. 

Comparisons of the WHO to the current Saudi growth charts showed a higher rate of stunting and wasting in Saudi children when the WHO charts were used [37]. The authors concluded that the WHO charts were constructed in privileged populations and are not suitable for use in some populations. Ideally, optimum growth should be based on well-nourished children and not unprivileged or undernourished children. Comparisons of the CDC to the current Saudi growth charts showed an underestimation of overweight and obesity, as well as an overestimation of undernutrition, stunting and wasting when the CDC charts were used [38,39]. The study that constructed the charts for BMI for age in 2009 aligned the charts to the CDC and WHO charts, but it is not clear how the prevalence of underweight/overweight were impacted [38].

## 4. Results: Anthropometrics Assessed in Saudi Children/Adolescents

### 4.1. Non-Traditional Measurements

There are body measurements that have been correlated with children’s growth status such as kidney length, penile size, body surface area and placental weight. Some studies explored these measurements in Saudi children including kidney length [40], penile size [41], body surface area [42], placental weight [43] hand grip strength [44]. These studies were not included in Table 2 and are summarized here. Mohtasib et al. reported that, in 950 Saudi children, left kidneys were longer than right kidneys, and consistent difference in kidney length by sex was observed. Both kidneys were longer in males than females. In addition, from several anthropometrics, height had the most significant correlation with kidney length [40]. Furthermore, AlHerbish established standards for penile size for normal full-term Saudi newborns. The mean penile length of 3.55 cm in this study was similar to previously reported international data [41]. The body surface area is of great interest for clinicians because it is widely used for determining drug dosages and for calculating the needs of patients for parenteral fluids and electrolytes [45]. From Saudi newborns in Abha, two simplified equations based on weight and height were created to calculate body surface area in Saudi newborns [42] and in adults [46].

### 4.2. Assessment of Growth Status in Studies Targeting Saudi Children

Table 2 shows a total of 167 studies assessing anthropometrics in Saudi children and adolescents. Table 2 reports the cutoffs used in the studies and whether the studies mentioned how the anthropometrics were measured. Studies assessing children’s growth using anthropometrics started since 1977 [47] (Appendix A) and included the capital city Riyadh and other cities, whereas other studies assessed children’s anthropometrics in villages, including Wadi Turaba [48], Khulais [49] and other villages [50]. Alfrayh and colleagues were the first to publish data from Riyadh and constructed physical growth standards [51].

**Table 2 healthcare-11-01010-t002:** Saudi children/adolescent studies measuring anthropometrics.

No.	Author, Year (Reference)	Regions	Population Age	Sample *n*	Anthropometrics Assessed	Anthropometric Assessment Definition (e.g., WHO, CDC, NCHS…)	Comments
Children Studies							
**1.**	Attallah et al., 1990 [52]	Asir	0–24 months	4520	Supine length, weight and head circumference	Compared to Wadi Turaba infants and to Europeans	No access to the paper, just the abstract. The need for national growth standards to assess the growth status of Saudi children was highlighted
**2.**	Wong and Al-Frayh 1990 [53]	Riyadh	Preschool children	6623	Weight, height, head and chest circumference and triceps skinfold	Not mentioned	Anthropometric measurement method mentioned
**3**	Al-Hazzaa 1990 [54]	Riyadh	6–14 years	1169	Height, weight, grip strength, chest, triceps, subscapular skinfold thickness	NCHS	Anthropometric measurement method mentioned. Figures of comparison were conducted between Saudi children and American and British populations
**4.**	Al-Omair 1991 [55]	Riyadh	Newborns	4498	Weight, height, head circumference	Not mentioned	Anthropometric measurement method not mentioned
**5.**	Al-Othaimeen 1991 [56]	10 cities	Birth– 90 years	933	Weight, height, weight for height, height for age	NCHS	PhD thesis.
**6.**	Al-sekait et al., 1992 [57]	All regions	School children		Weight, height, height for age, weight by age	NCHS	No access to the paper, just the abstract
**7.**	**Jan 1992** [58]	**Jeddah**	**Newborns**	**325**	**Weight, supine length, fronto-occipital head circumference, chest circumference, triceps skinfold thickness**	**Compared to national and international populations**	**Anthropometric measurement method mentioned. Normal anthropometric standards are presented for Saudi newborns born at sea level (Jeddah)**
**8.**	Magbool et al., 1993 [59]	Dammam, Al-Khobar, Qatif and Al-Hassa	6–16 years	21,638	Weight, height	NCHS	Anthropometric measurement method mentioned
**9.**	**Abolfotouh et al., 1993** [60]	**Aseer**	**1–60 months**	**1168**	**Weight, height**	**NCHS**	**The study adjusted the international growth curves for local use in Saudi preschool children**
**10.**	Kordy M 1993 [61]	Jeddah	1–18	3286	Weight, height and mid-arm circumference	Compared to European children	Saudi children are shorter in height and lighter in weight than European children
**11.**	Alfrayh and Bamgboye 1993 [62]	Riyadh	0–5 years	3795	Weight, height, weight for height, weight for age, height for age	NCHS/CDC	Saudi Arabian children are slightly shorter and thinner than their American counterparts
**12.**	**Alfrayh et al., 1993** [63]	**Riyadh**	**0–5 years**	**3795**	**Weight, height**	**WHO**	**Anthropometric measurement methods mentioned. The standard physical growth chart for Saudi Arabian preschool children was designed**
**13.**	**Chung 1994** [64]	**Dammam, Al-Khobar**	**6–16 years**	**21,638**	**Weight, height**	**NCHS**	**A microcomputer program for predicting percentile of height and weight by age for Saudi and US children aged 6–16 years was designed based on Magbool et al. data** [59]
**14.**	Al-fawaz et al., 1994 [65]	Riyadh	6–24 months	400	Weight, height, height for age, weight for height, weight for age	Compared to American reference population	Anthropometric measurement method mentioned. The WHO software ‘ANTHRO’ was used.
**15.**	Al-Eissa et al., 1995 [66]	Riyadh	infants	4578	Birth weight was collected within 24 h after birth	Normal birthweight-controls	Referenced the anthropometric measurement method
**16.**	Madani et al., 1995 [67]	Taif	infants	952	Weight, height, arm circumference, skinfold thickness	Control infants >2500 g	Anthropometric measurement method not mentioned
**17.**	**Al-Nuaim et al., 1996** [68]	**All regions**	**6- 18 years**	**9061**	**Weight, height, BMI**	**NCHS/CDC**	**No access to the paper, just the abstract. Growth charts for males 6–18 y old were created**
**18.**	Lawoyin 1997 [69]	Tabuk	infants	528	Birth weight	Compared to controls	Anthropometric measurement method mentioned
**19.**	**Al-Nuaim and Bamgboye 1998** [70]	**5 regions**	**6–11 years**	**4154**	**Weight and height, weight for height, height for age and weight for age**	**NCHS/CDC**	**Anthropometric measurement method mentioned. Growth charts for males 6–11 y old were created**
**20.**	**Al-Mazrou et al., 2000** [35]	**5 regions**	**0–5 years**	**24,000**	**Weight and height**	**NCHS**	**Anthropometric measurement method mentioned. Growth charts for children 0–5 years were created**
**21.**	Hashim and Moawed 2000 [71]	Riyadh	Newborns	500	Maternal anthropometrics and newborn weight	Compared to controls	Anthropometric measurement method not mentioned
**22.**	Alshammari et al., 2001 [72]	Riyadh	6–17 years	1848 children 2927 adults	Weight, Height, BMI	NHANES	Anthropometric measurement method mentioned
**23.**	Al-Hazzaa 2001 [73]	Riyadh	7–15 years	137	Weight, height, BMI, skinfolds, % body fat	Not mentioned	Anthropometric measurement method mentioned
**24.**	Al-Jassir et al., 2002 [74]	Riyadh	<5 years	21,507	Weight, Height	NCHS	Anthropometric measurement method mentioned
**25.**	El-Hazmi and Warsy 2002 [75]	5 regions	1–18 years	12,701	Weight, Height	Cutoffs of BMI based on Cole et al. [25]	Anthropometric measurement method mentioned
**26.**	**Al-Mazrou et al., 2003** [76]	**5 regions**	**0–5 years**	**23,821**	**Weight, height and head circumference**	**NCHS**	**The study compared the national growth monitoring data with NCHS growth standards, which was used in KSA. The study concluded “significant difference between the national growth monitoring data and the NCHS data, so it is important to use the national figures to avoid the drawbacks of NCHS standards”**
**27.**	Bamgboye and Al-Nahedh 2003 [77]	Northwestern	<3 years	332	Weight, height	NCHS/WHO	No access to the paper, just the abstract. The pattern of growth was negatively deviated compared to NCHS/WHO
**28.**	**Al-Amoud et al., 2004** [36]	**5 regions**	**0–5 years**	**23,821**	**Weight, height, head circumference**		**The study developed national growth charts from the national standards derived. Smoothed national growth standards with 5 and 7 percentiles were created and overcame the regional and the urban and rural variations**
**29.**	Al-Shehri et al., 2005 [78]	Abha and Baish	Newborns	5500	Birthweight, crown-to-heel length and head circumference	NCHS/CDC	Anthropometric measurement method mentioned. The anthropometry of newborns was less than that of the reference population.
**30.**	**Al-Shehri et al., 2005** [79]	**Abha**	**Newborns**	**6035**	**Birthweight, crown-to-heel length and head circumference**		**The study constructed intrauterine percentile growth curves for body weight, length and head circumference for local use in a high-altitude area of Saudi Arabia**
**31.**	**Al-Shehri et al., 2005** [80]	**Abha**	**0–24 months**	**5426**	**Weight, crown-toheel supine length, head circumference**	**NCHS**	**The study established growth reference standards for infants in the high-altitude Aseer region of southwestern Saudi Arabia. Abha infants in the present study were significantly smaller in all growth parameters than the NCHS**
**32.**	**Al-Shehri et al., 2006** [81]	**Abha**	**3–18 years**	**13,580**	**Weight, height, BMI**	**NCHS**	**The study standardized growth parameters for Saudi children (3–18 years) living at high altitude in Aseer region**
**33.**	Al-Saeed et al., 2006 [82]	Al-Khobar	6–17 years	2239	Weight, Height, BMI	Cutoffs of BMI based on Cole et al. [25] and the CDC	Anthropometric measurement method mentioned
**34.**	Abou-Zeid et al., 2006 [83]	Taif	School children from grade 1–6	465	Weight, height, weight for age, height for age and BMI for age	WHO/NCHS	Anthropometric measurement method mentioned
**35.**	Al-Rowaily et al., 2007 [84]	Riyadh	4–8 years	6207	Weight, height, BMI	NCHS	Anthropometric measurement method mentioned. Saudi children were more similar to Americans than to other Saudi children in different areas of Saudi Arabia
**36.**	Al-Hazzaa 2007 [85]	Jeddah	Preschool children	224	Weight, height, BMI, triceps and subscapular skinfolds, %fat, fat mass, fat free mass, FM index	Based on Slaughter et al. [86]	Anthropometric measurement method mentioned
**37.**	Al-Hazzaa 2007 [87]	Riyadh	6–14 years	1784	Weight, height, skinfold thickness, BMI, % fat, lean body mass	Based on several references	Anthropometric measurement method mentioned
**38.**	**El-Mouzan et al., 2007** [29]	**All regions,**	**birth–19 years**	**35,279**	**Weight, height, head circumference**	**Format of NCHS and CDC growth charts was adopted**	**The study established updated reference growth charts for Saudi children and adolescents. Anthropometric measurement method mentioned**
**39.**	**El-Mouzan et al., 2008** [38]	**All regions**	**birth–19 years** [29]	**35,279**	**Weight, height, head circumference**	**CDC growth charts**	**The study compared between the CDC and Saudi growth charts. There were major differences between the two growth charts. The study concluded “use of the 2000 CDC growth charts for Saudi children and adolescents increases the prevalence of undernutrition, stunting, and wasting”**
**40.**	Amin et al., 2008 [88]	Al-Hassa	Primary school children	1278	Weight, height, BMI	Cutoffs of BMI based on Cole et al. [25]	Anthropometric measurement method mentioned
**41.**	Alam 2008 [89]	Riyadh	female school children	1072	Weight, height, BMI	Cutoffs of BMI based on Cole et al. [25]	Anthropometric measurement method not mentioned
**42.**	**El-Mouzan et al., 2008** [90]	**All regions**	**children were from** [35]	**40,940**	**Weight, height**	**Compared growth data collected in 1994–1995 and 2004–2005**	**Anthropometric measurement methods were referred to in the studies. Evaluated the trend of nutritional status over 10 years. Improvement in Saudis’ nutritional status and a tendency toward overweight and obesity indicate the significance of growth chart update on regular basis**
**43.**	Al-Hashem 2008 [91]	Aseer	12–71 months	1041	Weight, height	WHO/NCHS	The study compared between PEM children in low and high-altitude regions. Anthropometric measurement method mentioned
**44.**	Khalid 2008 [92]	Southwest	6–15 years	912	Weight, height,	WHO/NCHS	Anthropometric measurement method mentioned
**45.**	**Al-Herbish et al., 2009** [30]	**All regions**	**0–19 years**	**35,275**	**Weight, height, BMI**	**WHO/CDC**	**The study established BMI curves with 10 percentiles that can be used for reference purposes for Saudi children and adolescents. In higher percentiles, Saudi children had equal or higher than Western children**
**46.**	El-Mouzan et al., 2009 [93]	[29]	birth–18 years	19,131	Weight, height, BMI, head circumference	Compared between regions	The study found significant differences in growth between regions of Saudi Arabia
**47.**	**El-Mouzan et al., 2009** [37]	**All regions**	**<5 years**	**15,516**	**Weight, height**	**A multinational sample selected by the WHO**	**WHO and Saudi growth standards were used for Saudi children and compared to each other. The study concluded “The use of the WHO standards in Saudi Arabia and possibly in other countries of similar socioeconomic status increases the prevalence of undernutrition, stunting, and wasting”.**
**48.**	El-Mouzan et al., 2010 [94]	All regions	<5 years	7390	Weight, height	WHO	Anthropometric measurement method mentioned [95]. The higher the education level of the head of the household, the lower the prevalence of malnutrition in their children
**49.**	El-Mouzan et al., 2010 [96]	All regions	<5 years	15,516	Weight, height	NCHS/WHO	Anthropometric measurement method mentioned. The study indicated significant regional disparities in prevalence of malnutrition in SA, with the highest in the southwestern region
**50.**	**El-Mouzan et al., 2010** [97]	**All regions** [29]	**<5 years**	**15,516**	**Weight, height**	**Saudi growth charts/WHO**	**Anthropometric measurement method mentioned. The study reported the prevalence of malnutrition in SA**
**51.**	Alwasel et al., 2011 [98]	Unizah	newborns	967	Weight, head circumference, chest circumference and body length at birth	the effect of Ramadan fasting on birth weight	Anthropometric measurement method not mentioned.
**52.**	Al-Hazzani et al., 2011 [99]	Riyadh	newborns	186	Weight	NICHD	Anthropometric measurement method not mentioned
**53.**	Warsy et al., 2011 [100]	Riyadh	newborns	151	Weight, height, BMI	Not mentioned	Anthropometric measurement method mentioned
**54.**	Batterjee et al., 2013 [101]	Makkah	6–15 years	1553	Height, head circumference	NCHS	Anthropometric measurement method not mentioned
**55.**	Wahabi et al., 2013 [102]	Riyadh	newborns	3426	Weight, length and head circumference	Not mentioned	Anthropometric measurement method not mentioned
**56.**	Wahabi et al., 2013 [103]	Riyadh	newborns	3231	Weight, length and head circumference	Not mentioned	Anthropometric measurement method not mentioned
**57.**	Bukhari 2013 [104]	Makkah	6–13 years	165	Weight, height, BMI	NHANES-II	Anthropometric measurement method mentioned
**58.**	Al-Saleh et al., 2014 [105]	Al-Kharj	newborns	1578	Weight, height, head circumference, crown-to-heel length	10th percentiles as cutoffs for dichotomizing birth anthropometric measures	Anthropometric measurement method not mentioned
**59.**	AlKarimi et al., 2014 [106]	Jeddah	6–8 years	417	Weight, height, height for age, weight for age, BMI for age	WHO	Anthropometric measurement method mentioned
**60.**	Alwasel et al., 2014 [107]	Baish	newborns	321	Weight, crown-to-heel length, circumferences of the head, chest and thigh	Associations between placental measurements and neonatal anthropometrics	Anthropometric measurement method mentioned
**61.**	Al-Shehri 2014 [108]	Makkah	6–12 years	258	Weight, height	Not mentioned	Anthropometric measurement method mentioned
**62.**	Munshi et al., 2014 [109]	Jeddah	infants	387	Weight, length, head circumference	Not mentioned	Anthropometric measurement method not mentioned
**63.**	Albuali 2014 [110]	Al-Ahsa	6–12 years	213	Weight, height, BMI, waist and hip circumference, waist-to-hip ratio	IOTF/CDC	Anthropometric measures followed the protocols of the International Society for the Advancement of Kinanthropometry
**64.**	Al-Mohaimeed et al., 2015 [111]	Al-Qassim	6–10 years	874	Weight, height, BMI, %body fat	WHO	Anthropometric measurement methods mentioned
**65.**	Al-Muhaimeed et al., 2015 [112]	Al-Qassim	6–10 years	874	Weight, height, BMI	Cole et al. [25]	Anthropometric measurement method mentioned
**66.**	**Shaik et al., 2016** [33]	**All regions**	**<5 years**	**15,601**	**Weight, height, BMI, head circumference**	**LMS (lambda, mu, sigma) methodology**	**The study produced growth references for Saudi preschool children**
**67.**	**El-Mouzan et al., 2016** [113]	**All regions**	**5–18 years**	**19,299**	**Weight, height**	**LMS methodology**	**The study produced growth reference for Saudi school-age children and adolescents**
**68.**	**El-Mouzan et al., 2016** [114]	**All regions**	**5–18 years**	**19,299**	**Weight for age, height for age, BMI for age**	**LMS and z-score reference**	**The study produced growth reference for Saudi school-age children and adolescents**
**69.**	Al-Qurashi et al., 2016 [115]	Al-Khobar	Newborns	476	Weight, length	CDC/WHO	Anthropometric measurement method not mentioned
**70.**	Farsi et al., 2016 [116]	Jeddah	7–10 years	914	Weight, height, BMI, waist circumference	Several references	Anthropometric measurement method mentioned
**71.**	Wyne et al., 2016 [117]	Riyadh	6–11 years	61	Weight, height, BMI	WHO/IOTF	Anthropometric measurement method mentioned
**72.**	Bhayat et al., 2016 [118]	Al-Madinah	12 years	419	Weight, height, BMI	WHO	Anthropometric measurement method mentioned
**73.**	Kensara and Azzeh 2016 [119]	Makkah	Infants	300	Weight, length and head circumference	Several references	Anthropometric measurement method mentioned
**74.**	AlKushi and Alsawy 2016 [120]	Makkah	Infants	200	Weight, length, head circumference	Cutoffs of birth weight with no reference	Anthropometric measurement method not mentioned
**75.**	Khalid et al., 2016 [121]	Aseer	Newborns	25	Weight, length, body circumferences and skinfold thicknesses	Compared newborn anthropometrics between low and high altitude	Anthropometric measurement method mentioned
**76.**	Eid et al., 2016 [122]	Jeddah	2–18 years	643	Birth weight, height	CDC and several references	Anthropometric measurement method not mentioned
**77.**	Bakhiet et al., 2016 [123]	Riyadh	6–12 years	1812	Height, head circumference	Not mentioned	Anthropometric measurement method mentioned
**78.**	**El-Mouzan et al., 2017** [34]	**All regions**	**0–60 months**	**15,601**	**Weight, height, head circumference**	**Compared Saudi Z-scores with WHO and CDC**	**The study established Z-score growth reference data for Saudi preschool children and growth charts**
**79.**	Quadri et al., 2017 [124]	Jazan	5–15 years	360	Weight, height, BMI	CDC	Anthropometric measurement method not mentioned
**80.**	AlSulaibikh et al., 2017 [125]	Dammam	7 days–13 years	527	Weight	determine the accuracy of the Broselow tape on estimating body weights	Anthropometric measurement method mentioned
**81.**	AlShammari et al., 2017 [126]	Hail	2–18 years	1420	Weight, height, BMI	WHO	Anthropometric measurement method mentioned
**82.**	Alsubaie 2017 [127]	Al-Baha	7–12 years	725	Weight, height, BMI	Not mentioned	Anthropometric measurement method not mentioned and not clear if it was self-report
**83.**	Al-Kutbe et al., 2017 [128]	Makkah	8–11 years	266	Weight, height, BMI, waist circumference	WHO	Anthropometric measurement method mentioned
**84.**	Saleh et al., 2017 [129]	Al-Ahsa	7–15 years	240	Weight, height, BMI	Saudi growth charts	Anthropometric measurement method mentioned
**85.**	Al-agha and Mahjoub 2018 [130]	Western	4–13 years	306	Weight, height, BMI	CDC/NHANES II	Anthropometric measurement method mentioned
**86.**	Belal et al., 2018 [131]	Taif	Newborn	1468	Weight, height, BMI, head circumference	Not mentioned	Anthropometric measurement method mentioned
**87.**	Sebiany et al., 2018 [132]	Dammam	6–12 years	851	Weight, height, mid-arm circumference and triceps skinfold thickness	Harvard standards and Saudi growth charts	Anthropometric measurement method not mentioned
**88.**	Shaban et al., 2018 [133]	Jazan	6–12 years	240	Weight, height, BMI	WHO	Anthropometric measurement method not mentioned
**89.**	Eldosouky et al., 2018 [134]	Al-Madinah	Children	294	Weight, height, BMI, waist and hip circumferences	WHO	Anthropometric measurement method mentioned
**90.**	Habibullah et al., 2018 [135]	Qassim	4–10 years	171	Weight, height, BMI	Not mentioned	Anthropometric measurement method mentioned
**91.**	Fakeeh et al., 2019 [136]	Jazan	6–13 years	300	Weight, height, BMI	WHO	Anthropometric measurement method mentioned
**92.**	**Mohtasib et al., 2019** [40]	**Riyadh**	**Newborns-14 years**	**950**	**Weight, height, BMI, body surface area**	**Saudi growth charts**	**Anthropometric measurement method not mentioned. The study established normal growth curves for renal length in relation to sex, age, body weight, height, BMI and body surface area of healthy children in Riyadh**
**93.**	Al-Hussaini et al., 2019 [137]	Riyadh	6–16 years	7930	Weight, height, BMI	WHO	Anthropometric measurement method mentioned
**94.**	Alahmari et al., 2019 [138]	Abha	6–16 years	200	Weight, height, BMI, hand dimensions	Not mentioned	Anthropometric measurement method mentioned
**95.**	Nasim et al., 2019 [139]	Riyadh	6–14 years	481	Weight, height, BMI	CDC	Anthropometric measurement method not mentioned
**96.**	Mosli R 2020 [140]	Jeddah	3–5 years	209	Weight, height, BMI Z-score	WHO	Anthropometric measurement method mentioned
**97.**	Alghadir et al., 2020 [141]	Not mentioned	8–18 years	500	Weight, height, BMI, waist circumference	Compared Saudis and expatriates	Anthropometric measurement method mentioned
**98.**	El-Gamal et al., 2020 [142]	Jeddah	Preschool	748	Weight, height, BMI,	WHO	Anthropometric measurement method not mentioned
**99.**	Alissa et al., 2020 [143]	Jeddah	5–15 years	200	Weight, height, BMI, waist circumference	IOTF	Anthropometric measurement method mentioned
**100.**	Alturki et al., 2020 [144]	Riyadh	9–12 years	1023	Weight, height, BMI, waist circumference	CDC	Anthropometric measurement method mentioned
**101.**	Gohal et al., 2020 [145]	Jazan	<5 years	440	Weight, height, BMI	WHO	Anthropometric measurement method mentioned
**102.**	Kamel et al., 2021 [146]	Hail	6–12 years	571	Weight, height, BMI, skinfold thickness	Based on references	Anthropometric measurement method mentioned
**103.**	Elsayed and Said 2021 [147]	Wadi aldawaser	10–12 years	150	Weight, height, arm circumference, BMI, Triceps skinfold	Jellife 1966 [148]	Anthropometric measurement method not mentioned
**104.**	Hijji et al., 2021 [149]	All regions	10–19 years	12,463	Weight, height, BMI	CDC	Anthropometric measurement method mentioned
**105.**	Gudipaneni et al., 2021 [150]	Aljouf	12–14 years	302	Weight, height, BMI, waist circumference	CDC	Anthropometric measurement method mentioned
**Adolescent Studies**							
**106.**	Abahussain N 1999 [151]	Al-Khobar	girls 12–19 years	676	Weight, height, BMI	NHANES-I	Anthropometric measurement method mentioned
**107.**	Abalkhail and Shawky 2002 [152]	Jeddah	10–19 years	2737	Weight, height, triceps skin fold thickness, mid-arm circumference, BMI	NHANES-I	Anthropometric measurement method mentioned
**108.**	Abalkhail et al., 2002 [153]	Jeddah	9–21 years	2860	Weight, height, BMI	NHANES-I	Anthropometric measurement method mentioned
**109.**	Al-Rukban 2003 [154]	Riyadh	Boys 12–20 years	894	Weight, height, BMI	NHANES	Anthropometric measurement method mentioned
**110.**	Al-Almaie 2005 [155]	Al-Khobar		1766	Weight, Height, BMI	NHANES and IOTF	Anthropometric measurement method mentioned
**111.**	**Al-Emran et al., 2007** [156]	**Riyadh**	**9–18 years**	**1053**	**Height**	**CDC/NHCS**	**The study provided growth reference values in body height and determined the specific age at peak height velocity for Saudi male and female adolescences in Riyadh. The study concluded that the use of CDC/NCHS height standards is not appropriate to be used in Saudi children**
**112.**	Farahat et al., 2007 [157]	Western, Northern and Eastern	12–19 years	1454	Weight, height, BMI	WHO/NCHS	Anthropometric measurement method not mentioned
**113.**	**Al-Hazzaa 2007** [158]	**All regions**	**Three studies** [57,68]		**Weight, height, BMI**	**BMI was plotted at the 50th and 90th percentiles**	**The study examined the trends in BMI of Saudi male adolescents between 1988 and 1996 from 3 nationally representative samples**
**114.**	Almuzaini 2007 [159]	Riyadh	11–19	44	Weight, height, BMI, Subscapular, triceps, thigh and calf skinfolds	Not available	Anthropometric measurement method mentioned
**115.**	Mahfouz et al., 2008 [160]	Abha	boys 11–19 years	2696	Weight, height, BMI	WHO	Anthropometric measurement method mentioned
**116.**	Bawazeer et al., 2009 [161]	Riyadh	adolescents	5877	Weight, height, BMI, waist-hip ratio	Based on a reference	Anthropometric measurement method mentioned
**117.**	El-Mouzan et al., 2010 [162]	Representative sample	0–19 years	35,279	Weight, height, head circumference, BMI	compared between Saudi males and females	Anthropometric measurement reference mentioned. The study determined the pattern and magnitude of differences in growth between boys and girls according to age
**118.**	El-Mouzan et al., 2010 [39]	Representative sample [29]	5–18 years	19,317	Weight, height, BMI	WHO/CDC	Anthropometric measurement method mentioned
**119.**	Al-Oboudi 2010 [163]	Riyadh	Girls 9–13 years	120	Weight, height, BMI	Not available	Anthropometric measurement method not mentioned
**120.**	Washi and Ageib 2010 [164]	Jeddah	13–18 years	239	Weight, height, skinfold thicknesses, BMI	CDC	Anthropometric measurement method mentioned
**121.**	**Al-Daghri et al., 2010** [165]	**Riyadh**	**5–17 years**	**964**	**Weight, height, waist, hip, sagittal abdominal diameter (SAD)**	**Cutoffs based on Cole et al.** [25]	**Anthropometric measurement method mentioned. The study established SAD cutoffs and their association with obesity**
**122.**	**Abahussain 2011** [166]	**Al-Khobar**	**15–19 years**	**721**	**Weight, height, BMI**	**NHANES-I**	**Anthropometric measurement method mentioned. The study determined the change in BMI among adolescent Saudi girls living in Al-Khobar between 1997 and 2007**
**123.**	**El-Mouzan et al., 2012** [167]	**3 regions, representative sample** [29]	**5–17 years**	**9018**	**Height**	**Compared stature between regions**	**Anthropometric measurement method: referred to the main study, a representative sample. The study assessed regional prevalence of short stature**
**124.**	Al-Attas et al., 2012 [168]	Riyadh	10–17 years	948	Weight, height, waist and hip circumference, BMI, Waist-to-hip ratio, SAD	Based on Cole et al. [25]	Anthropometric measurement method mentioned. The study concluded “The use of SAD may not be practical for use in the paediatric clinical setting”
**125.**	Al-Nakeeb et al., 2012 [169]	Al-Ahsa	15–17 years	1138	Weight, height, BMI	IOTF	Anthropometric measurement method mentioned; the study compared between Saudi and British anthropometrics
**126.**	**El-Mouzan et al., 2012** [170]	**3 regions**	**2–17 years**	**11,112**	**Weight, height, BMI**	**CDC**	**Anthropometric measurement method mentioned. The study assessed regional variation in prevalence of overweight and obesity**
**127.**	Al-Hazzaa et al., 2012 [171]	Al-Khobar, Jeddah, Riyadh	14–19 years	2906	Weight, height, BMI, waist circumference, waist/height ratio	IOTF	Anthropometric measurement method mentioned
**128.**	Al-Jaaly 2012 [172]	Jeddah	13–18 years	1519	Weight, height, BMI, waist circumference, waist-to-height-ratio	WHO	Anthropometric measurement method mentioned
**129.**	ALHazzaa et al., 2013 [173]	Riyadh and Al-Khobar	14–18 years	1648	Weight, height, BMI	IOTF	Anthropometric measurement method mentioned. The study compared between Saudi and British adolescents
**130.**	Al-Ghamdi 2013 [174]	Riyadh	9–14 years	397	Weight, height, BMI	IOTF	Anthropometric measurement method mentioned
**131.**	Al-Hazzzaa et al., 2014 [175]	Al-Khobar Jeddah Riyadh	14–19 years	2908	Weight, height, waist circumference, BMI	IOTF	Anthropometric measurement method mentioned
**132.**	Duncan et al., 2014 [176]	Al-Khobar and Riyadh	14–18 years	1648	Weight, height, BMI, waist circumference, waist to height ratio	IOTF	Anthropometric measurement method mentioned; Saudi anthropometric compared to British
**133.**	Al-Hazzaa 2014 [177]	Riyadh, Jeddah and Al-Khobar	15–19 years	2852	Weight, height, BMI, waist circumference	IOTF	Anthropometric measurement method mentioned
**134.**	**AlBuhairan et al., 2015** [178]	**All regions-representative sample**	**Saudi adolescents**	**12,575**	**Weight, height, BMI**	**CDC**	**Anthropometric measurement method mentioned. The national adolescent health study “Jeeluna” identified the health needs and status of adolescents in KSA.**
**135.**	Al-Daghri et al., 2015 [179]	Riyadh	12–17 years	1690	Weight, height, waist and hip circumference	Not mentioned	Anthropometric measurement method mentioned
**136.**	Al-Sobayel et al., 2015 [180]	Riyadh, Jeddah, Al-Khobar	14–19 years	2888	Weight, height, BMI, waist circumference	IOTF	Anthropometric measurement method mentioned
**137.**	Alenazi et al., 2015 [181]	Arar	Saudi male adolescents	523	Weight, height, BMI, waist circumference	CDC	Anthropometric measurement method mentioned
**138.**	Al-agha et al., 2015 [182]	Jeddah	6–14 years	586	Height	WHO	Anthropometric measurement method mentioned
**139.**	Shaik et al., 2016 [183]	Riyadh	9–16 years	304	Weight, height, BMI, waist and hip circumference	CDC/WHO	Anthropometric measurement method mentioned
**140.**	Moradi-Lakeh et al., 2016 [184]	All regions	15–25 years	2382	Weight, height, BMI	Not mentioned	Anthropometric measurement method not mentioned
**141.**	Al-agha et al., 2016 [185]	Jeddah	2–18 years	541	Weight, height, BMI	CDC	Anthropometric measurement method mentioned
**142.**	Hothan et al., 2016 [186]	Jeddah	11–18 years	401	Weight, height, BMI	CDC	Anthropometric measurement method not mentioned
**143.**	Al-Daghri et al., 2016 [187]	Riyadh	12–18 years	4549	Weight, height, BMI	Based on Cole et al. [25]	Anthropometric measurement method not mentioned
**144.**	Al-Agha et al., 2016 [188]	Jeddah	2–18 years	653	Weight, height, BMI	CDC	Anthropometric measurement method mentioned
**145.**	Alswat et al., 2017 [189]	Taif	12–18 years	424	Weight, height, BMI	WHO	Anthropometric measurement method not mentioned
**146.**	Omar et al., 2017 [190]	Taif	12–15 years	701	Weight, height, BMI, waist and hip circumferences, skinfold thickness, % body fat	IOTF	Anthropometric measurement method mentioned
**147.**	Nasreddine et al., 2018 [191]	All regions	Adolescents	1047	Weight, height, BMI, waist circumference	WHO/CDC	Anthropometric measurement method mentioned
**148.**	AlTurki et al., 2018 [192]	Riyadh	16–18 years	384	Weight, height, skinfold thickness and waist and hip circumferences	WHO	Anthropometric measurement method not mentioned
**149.**	Al-Hazzaa and Albawardi 2019 [193]	Riyadh, Jeddah, Al-Khobar	15–19 years	2888	Weight, height, BMI, waist circumference	IOTF	Anthropometric measurement method mentioned
**150.**	Fatima et al., 2019 [194]	Arar	15–19 years	322	Weight, height, BMI	CDC	Anthropometric measurement method mentioned
**151.**	Moukhyer et al., 2019 [195]	Jazan	12–19 years	502	Weight, height	WHO	Anthropometric measurement method mentioned
**152.**	Alowfi et al., 2021 [196]	Jeddah	12–19 years	172	Weight, height, BMI, waist circumference	Saudi growth charts	Anthropometric measurement method mentioned

**BOLD highlighted rows**: studies contributed to the establishment/adjustment of Saudi growth charts for children/adolescents or specific cutoffs or studied the trend of growth in representative samples or adjusted the international curves to be used in Saudis. Legend: BMI: Body mass index (may refer to z-scores of BMI), CDC: Centers for Disease Control, FM: Fat mass, IOTF: International Obesity Task Force, LBW: low birth weight, LMS: Lambda-Mu-Sigma, NCHS: National Center for Health Statistics, NHANESI: National Health and Nutrition Examinations Survey I, NICHD: National Institute of Child Health and Human Development, SA: Saudi Arabia, SAD, Sagittal abdominal diameter, WHO: World Health Organization

A total of 33 studies highlighted in Table 2 contributed to the establishment/adjustment of Saudi growth charts for children/adolescents or specific cutoffs or studied the trend of growth in representative samples or adjusted the international curves to be used in Saudis. Figure 2 shows the use of growth chart standards in Saudi children and adolescents. Figure 2 did not include studies that were used to establish growth standards at the national or regional levels. The majority of the studies used WHO (23%), CDC (17%), NCHS (17%) and IOTF (15%) cutoffs, although Saudi growth charts were published since 2005 [29]. Saudi growth charts are available on the website of the Saudi Ministry of Health (see Section 3.5); however, few studies used the Saudi growth chart cutoffs [34,40,97,129,132,196]. Only 4% of the studies used the Saudi growth charts (Figure 2). This is consistent with Mosli R. 2018 [197], who reported that 70% of pediatricians and dietitians in Saudi Arabi typically used CDC or WHO growth charts. However, the study included a small sample size (n = 105). Another study reported the low use of growth charts in the Middle East [198]. In regard to using the Saudi growth charts for Saudi adolescents, a study found that hospitals in Jeddah used international standards for anthropometrics (60%) compared to national standards of anthropometrics (10%). The study highlights the importance of standardizing the practice of anthropometric assessment among the Saudi adolescent group [199]. Larger representative studies are needed and identification of the reasons of not using the Saudi growth charts are crucial. Overall, most of the studies in children/adolescents were conducted in Riyadh and more studies are needed in other regions of Saudi Arabia. Few studies included a representative sample of the Saudi population from all regions (Table 2).

Discrepancies in classification of weight status have been observed in many studies when using different growth charts. A study in Spain reported that rates of overweight and obesity were doubled when plotted against the WHO growth standards compared to a locally established reference [200], similar in Iranian children [201]. Two studies in representative samples of Saudi children compared the children’s growth using the current Saudi growth charts, the WHO [37] and the CDC [38]. These studies found that the prevalence of undernutrition, stunting and wasting was increased in Saudi children when using the WHO standards or the 2000 CDC growth charts [37,38]. 

We explored the standards used in assessing children’s growth in Saudi Arabia across the years. Studies that were used to establish Saudi growth standards at the national or regional levels were excluded in this analysis. Figure 3 shows the breakdown of the used standards between 1980 and 2021 (A) and stratified into before 2008 and 2008 onwards (B). The year 2008 was selected as the current Saudi growth standards were endorsed by the Ministry of Health in 2007. In 2008 onwards, the WHO growth standards remained the main growth chart used (28%), followed by the 2000 CDC charts (21%); only 6% used the current Saudi growth charts. Before 2008, different versions of the National Center for Health Statistics charts were mainly used; almost no one used Saudi growth charts, although there were multiple published before the current Saudi charts. Issues in reporting were observed, such as not citing the source, unclear cutoffs/definitions and based on other references that are not commonly used. These differences in criteria used to define children’s growth create a challenge for many researchers and health officials as they cannot conclude whether the differences in overweight/obesity rates across regions and/or across the years are actual or are due to differences in the tools and definitions used.

## 5. Recommendations for Establishing Saudi Guidelines of Anthropometric Measurements and Directions for Future Work in Line with the Saudi 2030 Vision

This review did not focus on preterm infants; however, recommendations were provided based on the available evidence. Preterm infants’ growth should be monitored using postnatal growth standards specifically for preterm. These growth charts have been established using the INTERGROWTH-21st Project and can be used for the assessment of preterm infants until 64 weeks postmenstrual age [202]. These growth charts have been published in The Lancet Global Health Journal in 2015 and are generalizable to other populations if individuals are healthy, well-nourished and free from environmental and socioeconomic constraints on growth (Table 3). 

The Saudi growth charts have several limitations, including absent percentiles, no guidelines for interpretation and outdated cutoffs, which makes it necessary for an update. The WHO recommends that new estimates should be completed for anthropometric indicators every three years so that countries can regularly update their progress towards the Sustainable Development Goals (SDG goals) [15]. The endorsed growth charts on the website of the Saudi Ministry of Health were published in 2005. The WHO growth charts may be the recognized standard, particularly for children under 5 years old, as they are based on a prospective study using a sample of healthy children from six different countries on five different continents (Brazil, Ghana, India, Norway, Oman and the United States) with varying ethnic groups living in an environment that did not constrain optimal growth. However, only one country out of the twenty-three Arabic countries was included (Oman) and may not be similar to Saudi Arabia [203]. In addition, several countries had created their own growth charts and compared them with the WHO growth charts and reported misclassifications, including Korea [204], Spain [200] and Iran [201]. The United States adopted the WHO growth charts only for those aged 2 years and younger but still recommends the CDC charts for children older than 2 years. The rigorous methods used when creating the WHO growth standards for children 0–5 years are recommended when updating the Saudi growth charts. To prevent false impressions of a good percentage of the population being in a healthy condition due to an increase in the normal weight curves, several points and methods need to be considered depending on the age group (Appendix A). 

A complete guide providing recommendations for data collection, analysis and reporting on anthropometric indicators in children under 5 years old was published in 2019 by the WHO and UNICEF [15]. The guide is available in four languages, including an Arabic version, which is convenient for Saudi Arabia. The guide provides recommendations on planning, sampling, questionnaire development, training, standardization, equipment, field work processing and reporting data (Table 3). Several points need to be checked to assess the quality of anthropometric survey data to generate accurate child malnutrition status. These points are summarized in Section 3.1 in the WHO guidelines [15]. We recommend that, when updating the Saudi growth charts, they follow these guidelines to provide a clear understanding of the magnitude and distribution of malnutrition problems in the country and to design and monitor interventions to improve the nutritional status of the population concerned (Appendix A). However, it is important to note that the data quality cutoffs provided by the WHO for children < 5 years old need to be revised, as stated by the WHO, for various reasons [15]. One of the reasons was that these cutoffs were developed using a “set of surveys not all of which were nationally representative and included several rapid nutrition surveys conducted in emergency situations, where the populations concerned were probably more homogeneous with respect to nutrition status and its determinants”. Therefore, as a starting point, Saudi Arabia may develop a nationally representative survey using the accurate and rigorous guidelines provided by the WHO to ensure high-quality data that may help the WHO revise their data quality cutoffs. 

After updating the Saudi growth charts, endorsing them to be used by clinicians and publishing them on the Saudi MOH website are required. More studies are needed to identify the reasons of not using the Saudi growth charts (discussed in Section 4.2) [200,201]. To ensure that the Saudi growth charts are used, hospitals and clinics should include them in their protocols and reporting growth status should be documented in hospital systems. These data may help researchers to study growth trends and compare growth between regions [35,93,170], low and high altitudes [81], public and private schools and study factors impacting anthropometrics that may help in increasing life expectancy. In addition, the availability of anthropometrics data may assist in meeting the goals of the Kingdom’s Vision 2030 in the transformation towards a sustainable knowledge-based economy [1]. To use anthropometric data in research, it is recommended to conduct data cleaning of anthropometric data from electronic health records. A protocol has been proposed and applied in a large UK patient cohort [205]. A systematic review found several non-traditional anthropometrics that can be used to identify obesity in children [206]. These can be studied in the Saudi population and specific cutoffs developed to help in the prevention of childhood obesity. Overall, using the WHO/CDC or other growth charts results in over/underestimation of the prevalence of obesity, stunting, wasting and malnutrition [37,207]; thus, updating and using the Saudi growth charts are essential. 

After updating the Saudi growth charts and endorsing them to be used by all healthcare settings, specific cutoffs should be developed for 6–18-year-old children since the Saudi growth charts are up to 5 years only. Saudi growth charts were created for ≥6 years children–19 in 1996/1998 [68,70], and growth charts for the ages 5–19 years are available on the Saudi MOH website [208] from the study [29]. In 2015, a nationally representative adolescent study identified high prevalence of sedentary behaviours, poor dietary habits and overweight and obesity [178], indicating the need to update the Saudi growth charts for adolescents. The WHO published a guideline on school health services, such as referral and management of overweight and obesity [209]. A collaboration between the Saudi MOH and Saudi Ministry of Education is essential to develop a protocol for annual screening of anthropometrics of all children enrolled in governmental and private schools to ensure growth and prevent obesity and its comorbidities. Children in Saudi Arabia start Year 1 by the age of 6–7 years and kindergarten is not compulsory; therefore, an established protocol to ensure annual screening is necessary to include all children regardless of starting school (Table 3).

### Individuals with Health Conditions and Disabilities

A study showed that 718 newborns from 2000 to 2012 were registered with neural tube defects in King Faisal Specialist Hospital and Research Center. This was associated with not consuming folic acid preconception and in the first trimester [210]. Consequently, the creation of a national birth defect registry has been proposed to collect, analyse and report to policymakers [211]. These data may help in identifying their causes and conducting preventive measures. Currently, there are two national screening programs in Saudi Arabia: the premarital screening program and the national newborn screening program [212]. Both programs are mainly screening for genetic disorders in a manner either aiming to detect couples who are carriers or affected before their marriage or detecting affected newborns at early stages. The incidence of newborn errors in Saudi Arabia is one of the highest worldwide due to high rate of consanguineous marriages in Saudi Arabia. Compared to many countries, Saudi Arabia has only two screening programs and more screening programs are needed, such as the “newborn and infant physical examination” program conducted in the UK. 

Researchers created Saudi growth charts for individuals with Down syndrome in 2003 [213]. However, these charts need to be updated and distributed to be used and endorsed by the Saudi Ministry of Health. No studies have been found to assess the utilization of the Saudi growth charts for Down syndrome. Furthermore, body surface areas have been mentioned in Section 4.1 as they are a major factor in the determination of the course of treatment and drug dosage. Two studies were conducted in the Saudi population that created simplified equations based on anthropometric measurements to calculate body surface area in newborns and adults [43,47]. However, the studies included small sample sizes and larger representative studies are needed to create these equations. A recent study provided several equations of body surface area based on weight, height, sex and age of patients that can be used when creating these equations for the Saudi population [214]. A systematic review suggested that muscle mass may be the new body surface area for chemotherapy dosing [215]. 

**Table 3 healthcare-11-01010-t003:** Summary of recommendations for establishing Saudi guidelines of anthropometric measurements for children/adolescents and directions for future work in line with the Saudi 2030 Vision.

Recommendations—Children/Adolescents	Directions for Future Work
1. Use preterm infants’ growth standards for preterm infants if preterm infants are well-nourished and free from environmental and socioeconomic constraints on growth until 64 weeks postmenstrual age [202]	Develop a protocol to be used in all hospitals and to be endorsed by the Saudi MOH
2. Update the Saudi growth charts (SGC) using the WHO and the United Nations Children’s Fund (UNICEF) guideline for children under 5 years old, Arabic version [15] (see Appendix A)	▪ Use the WHO protocol for anthropometric measurement [15] to ensure standardization and provide a guideline for pediatricians, dietitians, general practitioners and family doctors on how to use the SGC and interpret them. ▪ Integrate SGC into electronic health records [216] Develop a protocol on anthropometric data documentation in hospital systems and to be reported to the Saudi MOH ▪ More studies on the assessment of growth status (growth charts used) of Saudi children and the reasons of why Saudi growth charts are not used by some clinicians are needed [200,201]
3. Explore non-traditional anthropometrics to define obesity in Saudi children and develop specific cutoffs [206]	Non-traditional anthropometrics, such as chest and wrist circumference, body mass abdominal index etc. [206] can be used in the case of limited equipment available and may help prevent childhood obesity by early detection
4. Update the SGC for children/adolescents ages 5–19 years [209] (see Appendix A)	▪ Collaboration between Saudi MOH and Saudi Ministry of Education to develop a protocol for annual screening of anthropometrics in schools [209] ▪ Provide a guideline for paediatricians, dietitians, general practitioners and family doctors on how to measure anthropometrics to ensure standardization and how to use the Saudi growth charts and interpret them [199] ▪ More studies on the assessment of growth status (growth charts used) of Saudi adolescents and the reasons of why Saudi growth charts are not used are needed [199] ▪ Develop a protocol on anthropometric data documentation in hospital systems and to be reported to the Saudi MOH
Recommendations for Individuals with Health Conditions and Disabilities	Directions for Future Work
5. Update SGC for Down syndrome [213]	Update, publish the Saudi cutoffs and endorse them to be used by the Saudi MOH
6. Create simplified equations to calculate body surface area in Saudi children and adults from large representative Saudi samples [43,47].	Equations to calculate body surface area are created from anthropometrics (e.g., weight, height). Body surface area is a major factor in the determination of the course of treatment and drug dosage and can be used by physicians in different medical conditions (e.g., transplantation, predict chances of survival in burn patients, nephrotic syndrome) [214]
	▪ Provide a guideline for health practitioners on when and how to use body surface equations to ensure standardization and how to interpret the results. ▪ Generalize and endorse the use of body surface equations by the Saudi MOH
7. Create a national birth defect registry [211]	The data from the registry will help in identifying the prevalence, trends, the causes of birth defects and conducting preventive measures.

Legend: MOH: Ministry of Health, SGC: Saudi growth charts, WHO: World Health Organization.

## 6. Conclusions

Anthropometric measurements are the first step in determining health status. In this review, we have summarized the evidence of anthropometric measurements used in Saudi children and adolescents. Saudi growth charts for children/adolescents are available; updating them and endorsing their utilization are essential to ensure standardization and help in determining health status. The Saudi growth charts are underused in research conducted on Saudi children and studies are needed to assess the reasons for not using them. This review provides recommendations for establishing Saudi guidelines of anthropometric measurements and directions for future work in line with the Saudi 2030 Vision. This review will help policymakers and the Saudi MOH to establish reports and national guidelines to be used in Saudi Arabia for anthropometric measurements and their valid cutoffs that may assist in detecting malnutrition and preventing it. 

## Figures and Tables

**Figure 1 healthcare-11-01010-f001:**
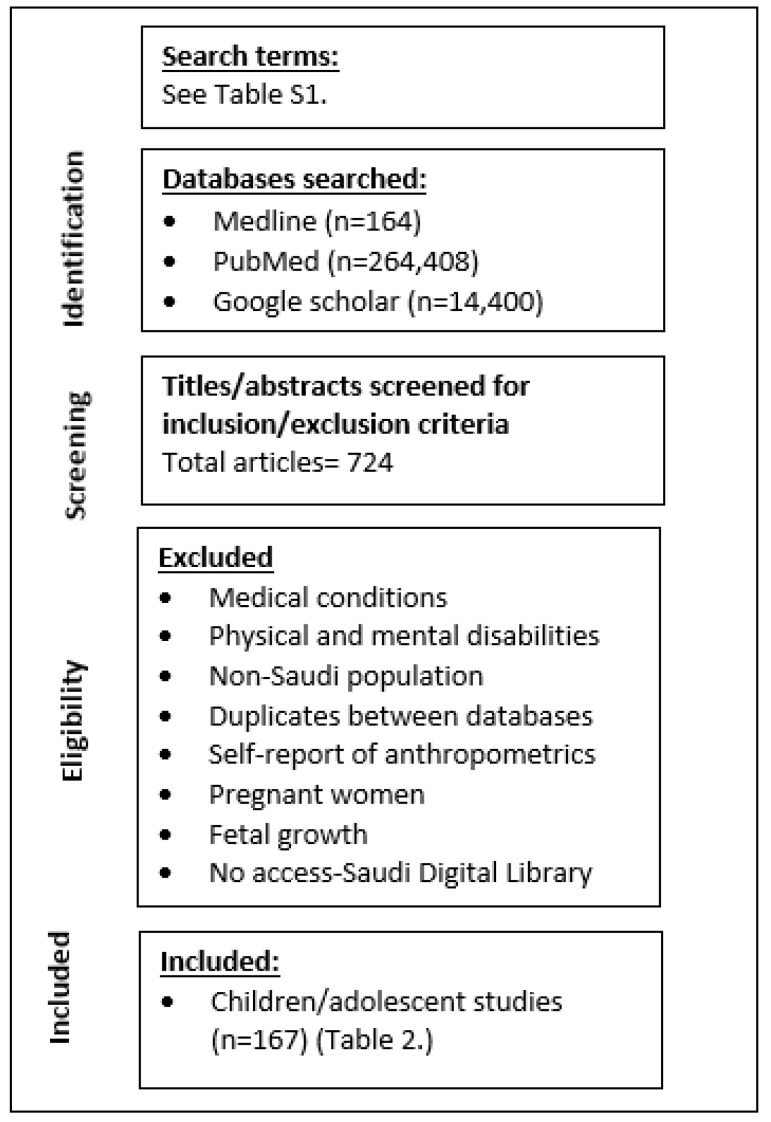
Search results (exclusion/inclusion) and total number of studies included.

**Figure 2 healthcare-11-01010-f002:**
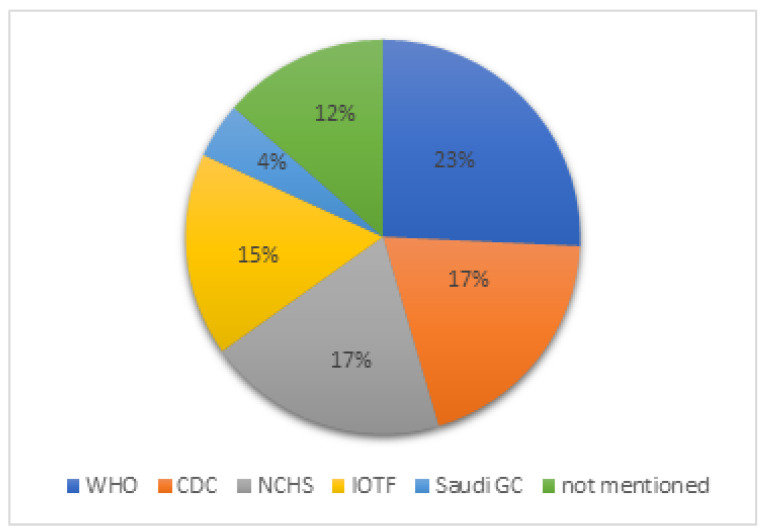
The use of growth standards in Saudi children and adolescents between 1980 and 2021. CDC: The US Centers for Disease Control, GC: growth charts, IOTF: The International Obesity Task Force, NCHS: National Center for Health Statistics, WHO: World Health Organization.

**Figure 3 healthcare-11-01010-f003:**
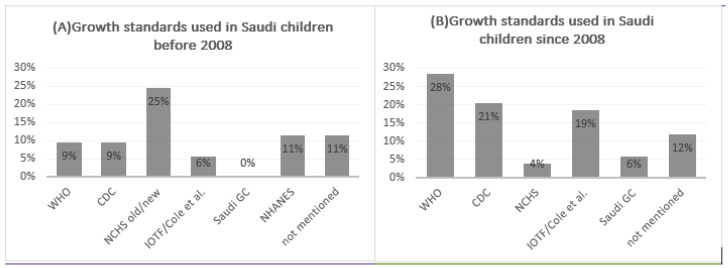
Growth standards used in research targeting Saudi children between 1980 and 2021. (**A**) Before 2008 and (**B**) since 2008. CDC: The US Centers for Disease Control, GC: growth charts, IOTF: The International Obesity Task Force, NCHS: National Center for Health Statistics, NHANES: National Health and Nutrition Examinations Survey, WHO: World Health Organization.

**Table 1 healthcare-11-01010-t001:** International and national growth standards used in Saudi children/adolescents.

WHO (birth–19 years)	Length/height for age Weight for age Head circumference velocity Weight for length/height Length velocity Head circumference for age Arm circumference for age Weight velocity Triceps skinfold for age Subscapular skinfold for age Motor development milestones Body mass index for age (BMI for age)
CDC (2–20 years) Recommends using the WHO growth charts for children ages 0 to 2 years	Length for age and Weight for age Head circumference for age and Weight for length (birth–36 months) Stature for age and Weight for age BMI for age (2–20 years) Weight for stature (2–5 years)
IOTF (2–18 years)	BMI cutoffs
Saudi Arabia (0–19 years)	Weight for age Length for age percentiles Head circumference for age Weight for length BMI for age

BMI: Body mass index, CDC: The US Centers for Disease Control, IOTF: The International Obesity Task Force, WHO: World Health Organization.

## Data Availability

No new data were created or analysed in this study. Data sharing is not applicable to this article.

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
