# Peer review of "Change in Growth Status and Obesity Rates among Saudi Children and Adolescents Is Partially Attributed to Discrepancies in Definitions Used: A Review of Anthropometric Measurements"

_healthcare, 2023, doi:10.3390/healthcare11071010_

Round 1

Reviewer 1 Report

Dear Authors, I carefully reviewed your paper. Please don't feel offended, but I think that your paper is not a valid one.

My first concerns is a methodological one: Why were only studies in English included? Are there no journals published in Arabic language, in this area? Since this is a paper focused on Saudi Arabia, the inclusion of researches published in Arabic language would have been a better choice, in my opinion, even a mandatory one.

But the real issues I have with your paper are here:

Lines 285-292: The fact that using local anthropometric standars may result in a different - often "more favourable" (i.e., helthier) - classification of children/adolescents does not mean that the use of those local standards should be encouraged. Local standards are likely to be more favourable if the general local population has higher BMI (or whatever measure is used), which does not mean that people in that country are healthy or "normal". This is why OMS standards should be used in any country, They are supposed to represent (and probably do) the vaste majority of world population. A consequence of using local standards is this: if for any reason (let say: bad or worsened eating habits) that population reaches higher weights, then also the curves of "normal weight" go up; if then "local BMI standards" are developped and applied, this could give the false impression that a good percentage of the population is in healthy conditions. The same (specular) reasoning, of course, applies also to situations of general undernutrition of an entire population.

Lines 330-331: The second part of this sentence is exactly the reason why an update of the Saudi growth charts not only is NOT essential, but on the contrary should be countered. Above all, it confirms my suspicions about a "getting worse" of food habits in your country, similarly to what happens in most wealthy countries. the simple "taking note" of this changes and theirs consequences, like an accountant does, does not improve doctors' and population's understanding of how bad this is for for the general health of  country.

These are the reasons why I consider your conclusions on this topic totally misleading, and a stroke of luck that local charts are still underused in Saudi reasearches, while your paper, if accepted, would lead to more efforts toward a bad (in my opinion) practice.

Author Response

Dear Authors, I carefully reviewed your paper. Please don't feel offended, but I think that your paper is not a valid one.

My first concerns are a methodological one: Why were only studies in English included? Are there no journals published in Arabic language, in this area? Since this is a paper focused on Saudi Arabia, the inclusion of researches published in Arabic language would have been a better choice, in my opinion, even a mandatory one.

We appreciate your comment. Researchers in health specialties publish their work in English journals as this is a requirement for promotion from all universities in Saudi Arabia. This is shown in table 2, studies conducted in Saudi Arabia measuring anthropometrics were published in English since 1981.

Other specialties such as Arabic Literature, Islamic Studies Education... etc. publish in Arabic and this is shown in the Saudi Digital Library that includes Arabic and English sources. The Arabic sources are mainly in the specialties mentioned earlier.

Saudi journals accepting health research, such as the Saudi Medical Journal, Journal of Taibah University Medical Sciences and other journals publish in English and the abstract only is in English and Arabic.

Therefore, this criteria (English language) was selected because the review is health-related and this research area publish only in English.

But the real issues I have with your paper are here:

Lines 285-292: The fact that using local anthropometric standards may result in a different - often "more favorable" (i.e., healthier) - classification of children/adolescents does not mean that the use of those local standards should be encouraged. Local standards are likely to be more favorable if the general local population has higher BMI (or whatever measure is used), which does not mean that people in that country are healthy or "normal". This is why OMS standards should be used in any country, they are supposed to represent (and probably do) the vast majority of world population. A consequence of using local standards is this: if for any reason (let say: bad or worsened eating habits) that population reaches higher weights, then also the curves of "normal weight" go up; if then "local BMI standards" are developed and applied, this could give the false impression that a good percentage of the population is in healthy conditions. The same (specular) reasoning, of course, applies also to situations of general undernutrition of an entire population.

Thank you for your comment. One of the aims for the Sustainable Development Goals (SDGs) and the six global nutrition targets for 2025 is to “end all forms of hunger and malnutrition by 2030”. Anthropometric data are collected to provide a clear understanding of the magnitude and distribution of malnutrition problems in a country, and to design and monitor interventions to improve the nutritional status of the populations concerned. Regarding the comment that the WHO growth charts should be used in any country because they represent the vast majority of the world population, only charts for children under 5 years were based on a sample of healthy children from six different countries on five different continents (Brazil, Ghana, India, Norway, Oman and the United States) with varying ethnic groups living in an environment that did not constrain optimal growth. Only one country out of the 23 Arabic countries was included (Oman) and may not be similar to Saudi Arabia [1]. This is not the case for children over 5 years, as the 2007 WHO growth charts for school-aged children and adolescents were established using a reconstruction of the 1977 NCHS/WHO growth reference [2]. The initial objective in the WHO report was to use 115 datasets from 45 countries, then this was narrowed down to 34 data sets from 22 countries. However, this was still challenging given the great heterogeneity in the methodology and other parameters.  They concluded “In consequence, WHO proceeded to reconstruct the 1977 NCHS/WHO growth reference from 5 to 19 years, using the original sample (a non-obese sample with expected heights), supplemented with data from the WHO Child Growth Standards (to facilitate a smooth transition at 5 years), and applying the state-of-the-art statistical methods” [2].

 In addition, several countries had created their growth charts and compared them with the WHO growth charts including Korea [3], Spain[4] and Iran[5]. We, however, believe the rigorous methods used by the WHO when creating their growth charts need to be applied when updating the Saudi Growth charts.

To clarify this to the reader, the following paragraph has been added to the discussion section (see lines 337-357):

Lines 330-331: The second part of this sentence is exactly the reason why an update of the Saudi growth charts not only is NOT essential, but on the contrary should be countered. Above all, it confirms my suspicions about a "getting worse" of food habits in your country, similarly to what happens in most wealthy countries. the simple "taking note" of this changes and theirs consequences, like an accountant does, does not improve doctors' and population's understanding of how bad this is for the general health of country.

These are the reasons why I consider your conclusions on this topic totally misleading, and a stroke of luck that local charts are still underused in Saudi researchers, while your paper, if accepted, would lead to more efforts toward a bad (in my opinion) practice.

The Saudi Growth charts have several limitations including absent percentiles, no guidelines for interpretation and outdated cutoffs which makes it necessary for an update. The WHO recommends that new estimates should be made for anthropometric indicators every three years so that countries can regularly update their progress towards the Sustainable Development Goals (SDG goals) [6]. The endorsed growth charts on the website of the Saudi Ministry of Health were published in 2005.This paragraph has been added to the discussion section (see lines 337-357). More recent anthropometric surveys conducted in Saudi Arabia were in 2016 and 2017 that reported the LMS and percentiles of growth in Saudi children in preschool. Several representative Saudi studies indicated rising trends of BMI, and these have been provided in the review.  Although these studies as you referred indicate that the populations diet is “getting worse”, they show the necessity of updating the Saudi growth charts as recommended by the WHO every three years.

To prevent false impressions of good percentage of the population is in a healthy condition due to an increase in the normal weight curves, several points and methods need to be considered depending on the age group. This has been added in the discussion section (see lines 337-357) (see lines 363-377) (see supplementary table S3.)

Submission Date

09 February 2023

Date of this review

24 Feb 2023 11:36:11

Dr. Essra Noorwali

References:

  1. Ng, S.W.; Zaghloul, S.; Ali, H.I.; Harrison, G.; Popkin, B.M. The prevalence and trends of overweight, obesity and nutrition-related non-communicable diseases in the Arabian Gulf States. Obes. Rev. 2011, 12.
  2. de Onis, M. Development of a WHO growth reference for school-aged children and adolescents. Bull. World Health Organ. 2007, 85, 660–667, doi:10.2471/BLT.07.043497.
  3. Kim, J.H.; Yun, S.; Hwang, S.S.; Shim, J.O.; Chae, H.W.; Lee, Y.J.; Lee, J.H.; Kim, S.C.; Lim, D.; Yang, S.W.; et al. The 2017 Korean national growth charts for children and adolescents: Development, improve ment, and prospects. Korean J. Pediatr. 2018, 61.
  4. Pérez-Bermejo, M.; Alcalá-Dávalos, L.; Pérez-Murillo, J.; Legidos-García, M.E.; Murillo-Llorente, M.T. Are the Growth Standards of the World Health Organization Valid for Spanish Children? The SONEV Study. Front. Pediatr. 2021, 9, doi:10.3389/fped.2021.700748.
  5. Hosseinpanah, F.; Seyedhoseinpour, A.; Barzin, M.; Mahdavi, M.; Tasdighi, E.; Dehghan, P.; Momeni Moghaddam, A.; Azizi, F.; Valizadeh, M. Comparison analysis of childhood body mass index cut-offs in predicting adulthood carotid intima media thickness: Tehran lipid and glucose study. BMC Pediatr. 2021, 21, doi:10.1186/s12887-021-02963-y.
  6. World Health Organization and the United Nations Children’s Fund (UNICEF) Recommendations for data collection, analysis and reporting on anthropometric indicators in children under 5 years old. Geneva: World Health Organization and the United Nations Children’s Fund (UNICEF); 2019;

Reviewer 2 Report

I would like to thank the authors for this interesting paper. I have the following suggestions:

Line 61: a diagnostic criterion, not criteria

Line 97: 2) The review will... There is no connection with the initial part of the senctence

Table 1: Why WHO data is in bold?

Line 163-164: The sentence needs to be rephrased

Line 200: 35,279 (no gap after the coma)

Line 201-203: the sentence needs to be rephrased

Table 2: Why some of the papers are in bold and others aren't?

And one more general remark: since during the last couple of decades there is a obesity of pandemic, many experts suggest that it is not correct to update each countries "normal" growth percentiles because obesity will be the new normal. Do you think that this approach affects the credibility of your paper? I think you should have it discussed in the Discussion Section.

Author Response

I would like to thank the authors for this interesting paper. I have the following suggestions:

Line 61: a diagnostic criterion, not criteria

This has been edited (see line 61)

Line 97: 2) The review will... There is no connection with the initial part of the sentence.

This has been edited (see lines 95-101)

Table 1: Why WHO data is in bold?

This has been removed (see table 1)

Line 163-164: The sentence needs to be rephrased.

This has been rephrased (see lines 163-165)

Line 200: 35,279 (no gap after the coma)

The gap has been removed (see line 201)

Line 201-203: the sentence needs to be rephrased.

This has been rephrased (see lines (203-206)

Table 2: Why some of the papers are in bold and others aren't?

These are studies that contributed to the establishment/adjustment of Saudi growth charts for children/adolescents or specific cut-offs or studied the trend of growth in representative samples or adjusted the international curves to be used in Saudis. The explanation has been added below table 2.

And one more general remark: since during the last couple of decades there is a obesity of pandemic, many experts suggest that it is not correct to update each countries "normal" growth percentiles because obesity will be the new normal. Do you think that this approach affects the credibility of your paper? I think you should have it discussed in the Discussion Section.

Thank you for this valuable comment. In response, this has been discussed in the discussion (see lines 337-357) (see lines 363-378) (see supplementary table S3.)

Reviewer 3 Report

The paper is a timely review.  It is generally well written although in places is a little repetitive, e.g.  paragraph 1 of the Introduction - ("Determinants of life expectancy trends is dependent on the “cardiovascular revolution” which combines smoking prevalence, obesity, lifestyle and related policies. Saudi Arabia is in the “cardiovascular revolution” due to the behavior-related problems of smoking, obesity and unhealthy lifestyle which should be addressed in order to increase life expectancy at birth [2]")

I suggest that the authors review the manuscript carefully to see if the text could be refined.

Paragraphs 4 and 5 of the Introduction (lines 70 to 94). Two separate elements of the use of anthropometric measurements are not clearly differentiated. Firstly, there are the technical aspects of anthro. measurements, e.g. correct measurement protocols for waist circumference; these are population agnostic. Secondly there is the clinical interpretation of the obtained data; these may be population-specific, e.g. percentile curves. The authors need to make clear this distinction and what is or should be Saudi-specific. For example, line 98, "standardized protocols and national guidelines in anthropometric measurements for Saudi children" could be read as implying that the technical aspects of measurement methods may need to be different from internationally recognised methods. I do not believe that this is what the authors mean (see lines 117/118).

Please clarify.

Line 125/6 Please make sure that Height is shown as raised to the power 2, i.e m2 or m^2 NOT simply ????â„Ž?(?)2

Table 2 is a comprehensive listing of existing data. It includes some very old data, e.g., ref 48  in 1977. Although of historical interest, given nutritional change in the intervening decades (line 330), I question the relevance of these data to contemporary populations. I suggest that thee older studies are moved to a Table in supplementary data. I note that the literature search (line 103) was limited to post-1990 publications.

The conclusions drawn are appropriate and the recommendations made (section 5) are useful and it is hoped that they are considered by the appropriate state bodies.

Author Response

The paper is a timely review.  It is generally well written although in places is a little repetitive, e.g.  paragraph 1 of the Introduction - ("Determinants of life expectancy trends is dependent on the “cardiovascular revolution” which combines smoking prevalence, obesity, lifestyle, and related policies. Saudi Arabia is in the “cardiovascular revolution” due to the behavior-related problems of smoking, obesity and unhealthy lifestyle which should be addressed to increase life expectancy at birth [2]")

This has been edited (lines 37-41).

I suggest that the authors review the manuscript carefully to see if the text could be refined.

Paragraphs 4 and 5 of the Introduction (lines 70 to 94). Two separate elements of the use of anthropometric measurements are not clearly differentiated. Firstly, there are the technical aspects of anthro. measurements, e.g., correct measurement protocols for waist circumference; these are population agnostic. Secondly there is the clinical interpretation of the obtained data; these may be population-specific, e.g., percentile curves. The authors need to make clear this distinction and what is or should be Saudi-specific. For example, line 98, "standardized protocols and national guidelines in anthropometric measurements for Saudi children" could be read as implying that the technical aspects of measurement methods may need to be different from internationally recognized methods. I do not believe that this is what the authors mean (see lines 117/118).

Please clarify.

Thank you for this valuable comment. This has been clarified (lines 69-97).

Line 125/6 Please make sure that Height is shown as raised to the power 2, i.e m2 or m^2 NOT simply ????â„Ž?(?)2

This has been edited (lines 123-124).

Table 2 is a comprehensive listing of existing data. It includes some very old data, e.g., ref 48 in 1977. Although of historical interest, given nutritional change in the intervening decades (line 330), I question the relevance of these data to contemporary populations. I suggest that the older studies are moved to a Table in supplementary data. I note that the literature search (line 103) was limited to post-1990 publications.

Thank you for your comment. It is very old, and we want it to show the discrepancies in standards used since that time. We agree that moving the 80s studies to the supplementary materials is necessary therefore, we moved them to Table S2 in supplementary material.  In lines 100-102, it was mentioned that hand searches of reference lists of retrieved articles were undertaken and studies prior to 1990 were included. 

The conclusions drawn are appropriate and the recommendations made (section 5) are useful and it is hoped that they are considered by the appropriate state bodies.

Thank you for your comment. 

Round 2

Reviewer 1 Report

Thank you for your reply and for the changes you made to your article. I am now satisfied with the explanations and clarifications you added. Thank you for accepting my concerns. Best regards.

Author Response

Dear reviewer, 

Thank you for reviewing our manuscript. Your comments were valuable and have made our manuscript clearer to the readers and the section we added in the discussion was a significant section that was needed. 

Thank you again. 

Dr. Essra Noorwali